# Patterns and impacts of patient migration for primary knee arthroplasty in China: A national retrospective study

**Fanqiang Meng[1], Xinjia Deng[1], Junzhi Sheng[1], Haoran Feng[1], Liusong Shen[1], Hu Chen[2], Dongxing Xie[1,3,4,5], Huizhong Long** [1,3,4,5]*

1 Department of Orthopaedics, Xiangya Hospital, Central South University, Changsha, Hunan, China, 2 Tibet Autonomous Region People's Hospital, Lhasa, Tibet, China, 3 Hunan Key Laboratory of Joint Degeneration and Injury, Changsha, Hunan, China, 4 Key Laboratory of Aging-related Bone and Joint Diseases Prevention and Treatment, Ministry of Education, Xiangya Hospital, Central South University, Changsha, Hunan, China, 5 National Clinical Research Center for Geriatric Disorders, Xiangya Hospital, Central South University, Changsha, Hunan, China

☯ These authors contributed equally to this work.
* huizlong@csu.edu.cn

## Abstract

### Objective

The phenomenon of interregional patient migration for primary knee arthroplasty (KA) appeared due to unbalanced distribution of medical resources. However, there is a paucity of literature about migration among patients receiving primary KA and the possible influence of migration on postoperative outcomes and financial burden. This study was aimed to investigate the characteristics of patient migration for KA in China and evaluate the related impacts.

### Methods

Primary KAs performed between 2013 and 2018 were retrieved from the Hospital Quality Monitoring System in China. Cross-province (i.e., 31 provinces of mainland China) attendance for KA was assessed. Propensity score-matched analysis was conducted to evaluate the effect of migration on postoperative outcomes and financial burden.

### Results

A total of 168,693 primary KA patients were included during the study period. The migration rate decreased from 17.6% in 2013 to 7.6% in 2018 (P < 0.001). Migration occurred more frequently in patients with fewer comorbidities. Migrating patients tended to choose provincial hospitals and hospitals with high procedure volumes. Beijing and Shanghai were the most preferred destinations. After matching, there were no significant differences in in-hospital mortality, pulmonary embolism, deep

**Data availability statement:** The data that support the findings of this study are available from the corresponding author Huizhong Long upon reasonable request with permission of the Department of Medical Administration, National Health Commission of the People's Republic of China. To facilitate external data requests, we have designated the Ethics Committee of Xiangya Hospital as the point of contact. The Committee can be reached at xyyyllwyh@126.com.

**Funding:** This work was supported by the National Key Research and Development Plan (2022YFC3601900 and 2022YFC2505500), the National Natural Science Foundation of China (U21A20352 and 82372474), the China Postdoctoral Science Foundation (2023M743958 and 2025M772148), the Science and Technology Program of Hunan Province (2025JJ60479 and 2023JJ41003), the Central South University Innovation-Driven Research Programme (2023CXQD031), and the Fundamental Research Funds for the Central Universities of Central South University (2024ZZTS0179 and 2021ZZTS0355). The funders had no role in study design, data collection and analysis, decision to publish, or preparation of the manuscript.

**Competing interests:** The authors have declared that no competing interests exist.

vein thrombosis, wound infection, or 30-day readmission between two groups. The migration group had much higher total hospital charges.

## Conclusion

A noticeable number of patients required cross-province migration to get access to KA. Migration didn't make a big difference but caused a greater financial burden. It is imperative to take initiatives to reduce disparities in resource allocation instead of relying on cross-district access.

## Introduction

Knee arthroplasty (KA) is a safe and well-recognized musculoskeletal surgical procedure that is highly clinically effective for alleviating pain and improving both function and quality of life for patients with advanced knee arthritis [1]. More than 700,000 KA procedures have been performed annually in the United States, and that number has been increasing rapidly [2–4]. Given the high volume of these procedures, KA represents a substantial economic and quality of care burden on the healthcare system [4]. Moreover, patients in remote regions sometimes have to wait longer time or travel to receive KA due to limited professional medical resources [5].

Due to the interregional disparity in the distribution of healthcare resources, there appeared mobility of the population to take advantage of the health services of other regions. This phenomenon is called interregional patient migration [6]. Recently, this phenomenon has attracted the attention of the academic community, especially in terms of the quality of healthcare provided, the safety of patients, resource utilization, and the need to plan investments and the reallocation of medical resources. Several studies have reported this interregional migration among patients diagnosed with cancer, patients requiring revision arthroplasty, and liver transplantation candidates [7–10]. In the study by Lawson et al., over 38% of patients migrated for knee revision procedures, and teaching institutions were the preferred migration destination for approximately 78% of patients [8]. However, there is a paucity of literature about migration among patients receiving primary arthroplasty and the possible influence of migration on postoperative outcomes and financial burden.

There is growing attention to equity and patients' limited options in all healthcare practices. Assessing the sufficiency of healthcare supplies in undeveloped communities has long been a challenge for administrators and policymakers. It is vital to obtain epidemiological information and investigate the impact of patient migration to provide evidence-based references for formulating targeted policies to alleviate related medical and financial burdens. Even with a long travel and less reimbursement [11], there were a large number of patients traveling across borders for medical services in China due to inequities in the geographical distribution of healthcare resources [12], which provides an proper setting for the evaluation of patient migration. Therefore, in this study, using a large national database in China, we investigated the patterns

of migration among patients receiving primary KA and evaluated the effect of migration on postoperative outcomes and financial burden.

## Materials and methods

### Data source

The Hospital Quality Monitoring System (HQMS) database was accessed for this study [13–15]. The HQMS database is a mandatory patient-level national registration database of standardized electronic inpatient discharge records from tertiary hospitals, under the administration of the National Health Commission of the People's Republic of China. Tertiary hospitals in China have been mandated to automatically submit standardized inpatient discharge records daily to HQMS since January 1, 2013. The HQMS data reporting system performs daily automated quality control at the time of data submission to ensure the completeness, consistency, and accuracy of data [16].

### Ethics statement

This study adhered to all relevant policies concerning patient privacy. Under the authorization of the HQMS Committee Board, this work has been approved by the institutional ethical review board (reference number: 2017121016). There was no direct contact with patients or primary collection of individual patient data. All data were deidentified and the results of the study were provided in tabular form, ensuring that patient identification was omitted. Therefore, obtaining informed consent was not necessary.

### Study population

We utilized patient migration data for primary KA extracted from the HQMS database on October 16, 2023. All patients who underwent elective primary KA were identified using procedure codes according to the International Classification of Diseases, Ninth Revision (ICD-9; 81.54) from January 1, 2013 to December 31, 2018. We also searched other knee-related procedure codes and medical terms to identify patients. Diagnosis codes and text related to hospitalization were reviewed by two orthopedists independently to verify whether a primary KA had been performed, with disagreements resolved via discussion. Sociodemographic factors and clinical information were collected for each hospitalization. Patients with missing demographic data (i.e., sex, age, and residence) and undergoing KA for a non-elective indication were excluded. Overall, 168,693 patients undergoing primary KA were included in this study.

### Study variables

The exposure variable of this study was migration (defined as patients receiving KA in a hospital located in a province that was outside of the patient's habitual residence) versus non-migration. Characteristics identified as covariates included age, sex, year of surgery, marital status, indication for surgery (osteoarthritis or non-osteoarthritis), hospital type, hospital volume of procedure, and comorbidities (ICD-10 diagnostic code for each disease). Comorbidities included congestive heart failure, myocardial infarction, peripheral vascular disease, cerebrovascular disease, dementia, rheumatic disease, peptic ulcer, hemiplegia, chronic renal disease, tumors, liver disease, diabetes, coronary artery disease, hypertension, chronic obstructive pulmonary disease (COPD), and Charlson comorbidity index (CCI; calculated based on the recorded morbidities) [17].

### Outcomes

The outcomes of interest in this study included postoperative outcomes, and financial burden. Postoperative outcomes referred to in-hospital mortality, pulmonary embolism, deep vein thrombosis, wound infection, and 30-day readmission. Financial burden referred to total hospital charges.

## Statistical analysis

Patients were stratified by age, sex, marital status, indication for surgery, comorbidities, geographic region of residency, and hospital characteristics (volume of KA and hospital type). Continuous data were expressed as the mean ± standard deviation (SD), while categorical data were expressed as numbers and proportions (percentage). We divided the indications for KA into two categories: osteoarthritis and non-osteoarthritis (e.g., rheumatoid arthritis, osteonecrosis, and other knee disorders) [15]. The CCI was calculated and divided into 3 categories: 0, 1–2, and ≥3 [14]. The hospital volume of procedures was categorized into 3 groups: >500/year, 250–500/year, and <250/year [18]. The patient characteristics were compared between the migration and non-migration groups by using Student's t test for continuous variables and the chi-square test for categorical variables.

To control for confounding factors between the migration and non-migration groups, propensity score matching was adopted to match the migration patients to non-migration controls. In consideration of the secular trends in KA in relation to various covariates at different calendar times, matched patients were constructed within 1-year blocks of calendar time [19]. Propensity scores were calculated using logistic regression from patient characteristics and hospital characteristics for each eligible patient [19], after further excluding patients undergoing bilateral KA or with missing records of charges or readmission. By utilizing a greedy matching algorithm, each migrated patient was matched to a non-migrated patient in a 1:1 ratio by propensity score within each 1-year accrual block [20]. The confounding variables incorporated in the logistic model included age, sex, marital status, indication for surgery, CCI category, each specific comorbidity, geographical region (North, East, North-East, South-Central, South-West, and North-West regions) [15], hospital type, and hospital volume. The standardized mean difference for each variable between the migration group and non-migration group was computed to measure covariate balance before and after matching [21]. A standardized mean difference (SMD) of ≤0.1 was considered suggestive of covariate balance.

The total hospital charges were analyzed as continuous variables, and they were also used as dichotomous variables. i.e., increased total hospital charges were defined as higher than the 75th percentile. The total hospital charges were processed by lognormal transformation to create more normal distributions. Linear regression was used to calculate the parameter estimates for total hospital charges, while the coefficients were converted to odds ratios (ORs) and the 95% confidence interval (CI) through exponential transformation and described as percentage differences [22,23]. The effects of migration on deep vein thrombosis, pulmonary embolism, wound infection, 30-day readmission, and increased total hospital charges were analyzed by logistic regression. Poisson regression was applied to analyze the effect of migration on in-hospital mortality [24]. Asymmetric propensity score trimming was performed using a greedy matching algorithm to exclude patients with a propensity score <2.5% based on its distribution in the migration group and >97.5% based on its distribution in the non-migration group to eliminate the potential impact of unmeasured confounding [25].

All statistical analyses were performed using SAS 9.4 (SAS Institute Inc., Cary, NC, USA), and a P value <0.05 was considered statistically significant.

## Results

A total of 168,693 patients who underwent primary KA were identified during the period between 2013 and 2018 in HQMS (Fig 1). Of these, there were 19,511 migrated patients. The overall migration rate for patients receiving KA was 11.6%. A summary of the characteristics of the patients in the two groups is shown in Table 1. There was an apparent downward trend in the proportion of migrated patients in all KA procedures throughout the 6 years from 2013 to 2018 (P<0.001) (Fig 2).

Compared with non-migrated patients, migrated patients were slightly more female and married. In addition, migration occurred more frequently in patients with fewer comorbidities. Migrated patients had a lower CCI, and the proportion of patients with several comorbidities was also lower in migrated patients. In terms of hospital characteristics, there were

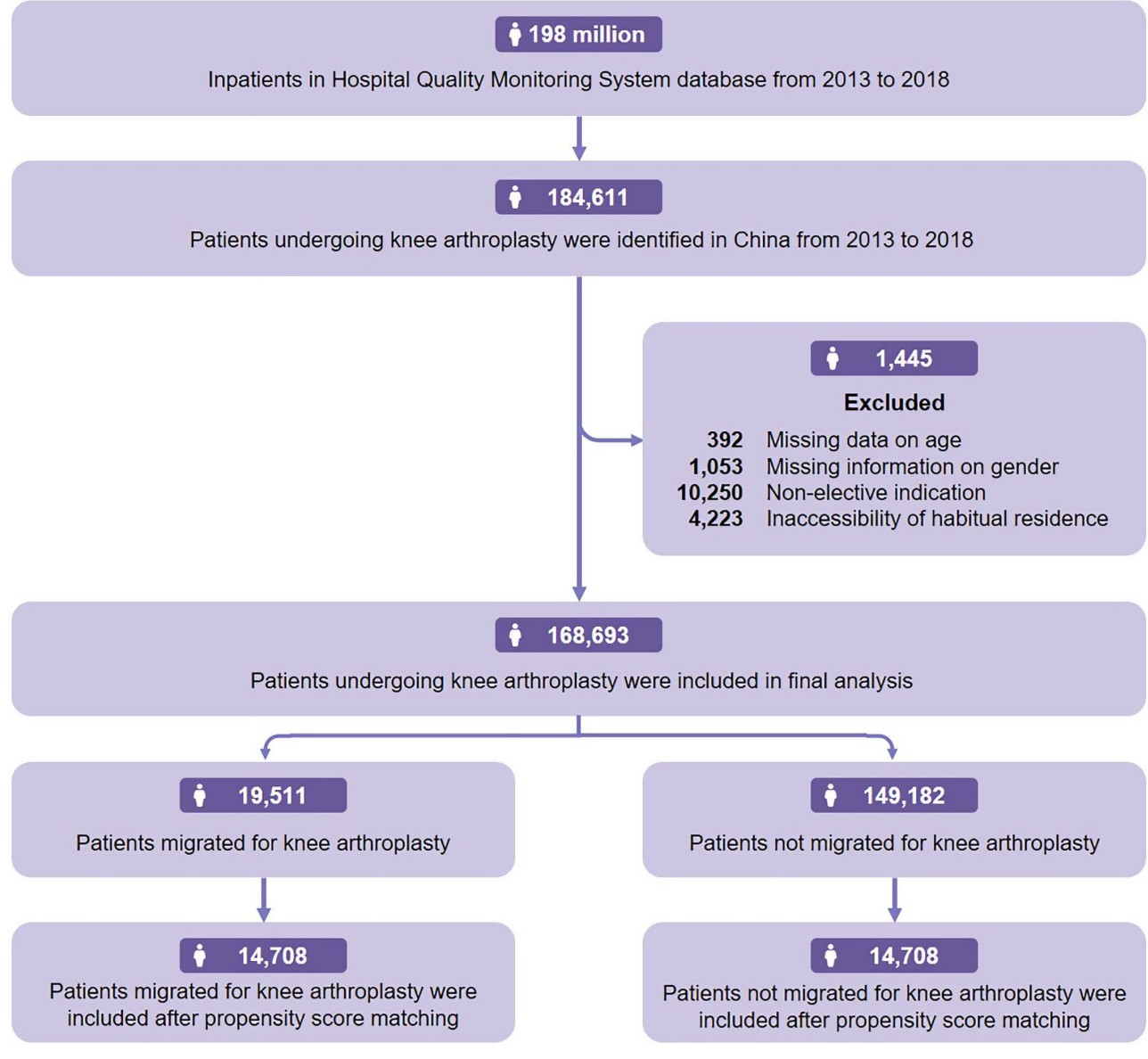

**Fig 1. Flowchart of analysis.**

also apparent differences between the two groups. Migrated patients tended to choose provincial hospitals and hospitals with higher volumes than their counterparts (Table 1).

As shown in Fig 3, a large proportion of patients from northern provinces chose to migrate for KA. In relatively high-income provinces, a high proportion of patients migrated from surrounding provinces. Beijing and Shanghai were the most preferred migration destinations, with 44.2% and 34.2% of patients migrating from outside provinces, respectively (Fig 4). However, the distribution of patient migration for primary KA in different provinces varied (Fig 5). Generally, patients tended to undergo KA in high-income provinces that are close to their usual residence. For instance, the

**Table 1.** Demographics of patients who underwent primary knee arthroplasty during 2013 to 2018.

| | All patients N = 168,693 | Not migrated N = 149,182 | Migrated N = 19,511 | P value |
|---|---|---|---|---|
| **Age, mean (SD), years** | 66.6 (8.2) | 66.8 (9.1) | 65.0 (8.4) | <0.001 |
| **Year (%)** | | | | <0.001 |
| 2013 | 18,458 (10.9) | 15,208 (10.2) | 3250 (16.7) | |
| 2014 | 25,684 (15.2) | 22,199 (14.9) | 3485 (17.9) | |
| 2015 | 28,558 (16.9) | 24,853 (16.7) | 3705 (19.0) | |
| 2016 | 31,638 (18.8) | 27,807 (18.6) | 3831 (19.6) | |
| 2017 | 31,421 (18.6) | 28,694 (19.2) | 2727 (14.0) | |
| 2018 | 32,934 (19.5) | 30,421 (20.4) | 2513 (12.9) | |
| **Sex (%)** | | | | 0.016 |
| Female | 131,719 (78.1) | 116,353 (78.0) | 15,366 (78.8) | |
| Male | 36,974 (21.9) | 32,829 (22.0) | 4145 (21.2) | |
| **Marital status (%)** | | | | <0.001 |
| Married | 155,624 (92.2) | 137,450 (92.1) | 18,174 (93.2) | |
| Single | 8424 (5.0) | 7449 (5.0) | 975 (5.0) | |
| Unknown | 4645 (2.8) | 4283 (2.9) | 362 (1.9) | |
| **Indication (%)** | | | | <0.001 |
| Osteoarthritis | 157,972 (93.6) | 139,673 (93.6) | 18,299 (93.8) | |
| Non-osteoarthritis | 10,721 (6.4) | 9509 (6.4) | 1212 (6.2) | |
| **Charlson Comorbidity Index (%)** | | | | <0.001 |
| 0 | 127,950 (75.9) | 112,075 (75.1) | 15,875 (81.4) | |
| 1-2 | 38,697 (22.9) | 35,188 (23.6) | 3509 (18.0) | |
| ≥3 | 2046 (1.2) | 1919 (1.3) | 127 (0.7) | |
| **Comorbidity (%)** | | | | |
| Myocardial infarction | 252 (0.2) | 232 (0.2) | 20 (0.1) | 0.071 |
| Congestive heart failure | 1292 (0.8) | 1223 (0.8) | 69 (0.4) | <0.001 |
| Peripheral vascular disease | 4077 (2.4) | 3838 (2.6) | 239 (1.2) | <0.001 |
| Cerebrovascular disease | 7034 (4.2) | 6485 (4.4) | 549 (2.8) | <0.001 |
| Dementia | 488 (0.3) | 469 (0.3) | 19 (0.1) | <0.001 |
| Rheumatic disease | 5687 (3.4) | 4945 (3.3) | 742 (3.8) | <0.001 |
| Peptic ulcer | 727 (0.4) | 649 (0.4) | 78 (0.4) | 0.480 |
| Hemiplegia | 38 (0.0) | 36 (0.0) | 2 (0.0) | 0.224 |
| Chronic renal disease | 555 (0.3) | 522 (0.4) | 33 (0.2) | <0.001 |
| Tumor | 1074 (0.6) | 993 (0.7) | 81 (0.4) | <0.001 |
| Liver disease | 4551 (2.7) | 4207 (2.8) | 344 (1.8) | <0.001 |
| Diabetes | 20,849 (12.4) | 18,987 (12.7) | 1862 (9.5) | <0.001 |
| Coronary artery disease | 5169 (3.1) | 4885 (3.3) | 284 (1.5) | <0.001 |
| Hypertension | 65,349 (38.7) | 59,245 (39.7) | 6104 (31.3) | <0.001 |
| COPD | 2185 (1.3) | 2058 (1.4) | 127 (0.7) | <0.001 |
| **Geographical region of residency (%)** | | | | <0.001 |
| North | 33,829 (20.1) | 27,088 (18.2) | 9302 (47.7) | |
| East | 67,796 (40.2) | 61,304 (41.1) | 6492 (33.3) | |
| North-East | 8105 (4.8) | 6668 (4.5) | 1437 (7.4) | |
| South-Central | 32,764 (19.4) | 30,618 (20.5) | 2146 (11.0) | |
| South-West | 13,174 (7.8) | 11,830 (7.9) | 1344 (6.9) | |
| North-West | 13,025 (7.7) | 11,674 (7.8) | 1351 (6.9) | |

*(Continued)*

**Table 1.** (Continued)

| | All patients N = 168,693 | Not migrated N = 149,182 | Migrated N = 19,511 | P value |
|---|---|---|---|---|
| **Type of hospital (%)** | | | | <0.001 |
| Provincial | 104,368 (61.9) | 88,052 (59.0) | 16,316 (83.6) | |
| Non-provincial | 64,325 (38.1) | 61,130 (41.0) | 3195 (16.4) | |
| **Hospital volume (%)** | | | | <0.001 |
| >500/year | 39,926 (23.7) | 28,839 (19.3) | 11,087 (56.8) | |
| 250–500/year | 34,705 (20.6) | 31,793 (21.3) | 2912 (14.9) | |
| <250/year | 94,062 (55.8) | 88,550 (59.4) | 5512 (28.3) | |

COPD, chronic obstructive pulmonary disease.

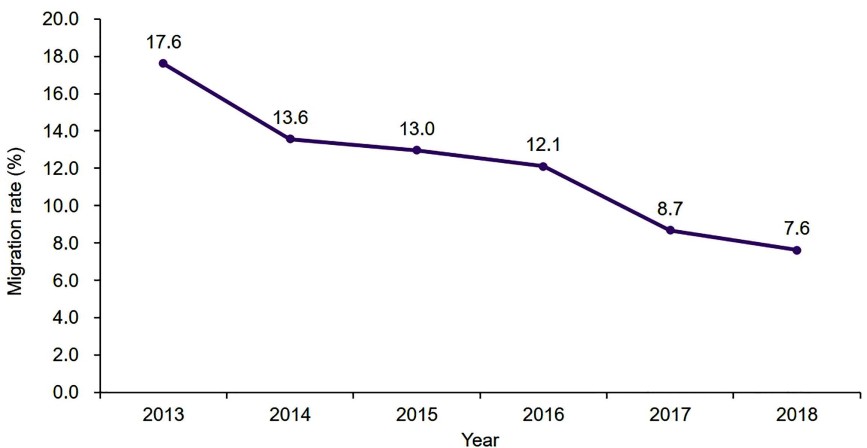

**Fig 2. Patient migration rates for primary knee arthroplasty from 2013 to 2018.**

proportion of arthroplasty performed in Jiangsu was the highest among Anhui patients who seek hospitalization in other provinces, while most patients residing in Hebei chose to receive primary KA in Beijing.

After propensity score matching, 14,708 pairs (mean age [SD], 65.8 [8.3] years; 78.7% were women) were included, and the baseline characteristics between matched groups were well balanced, with all standardized mean differences <0.1 (Table 2). Compared to non-migration, there were no significant associations between migration and in-hospital mortality (OR 1.00; 95%CI: 0.20–4.95), pulmonary embolism (OR 1.25; 95%CI: 0.59–2.67), deep vein thrombosis (OR 1.08; 95%CI: 0.88–1.34), wound infection (OR 1.10; 95%CI: 0.60–2.02) or 30-day readmission (OR 0.89; 95%CI: 0.75–1.05) (Table 3).

The comparison of financial burden is shown in Table 3. The migration group had greater total hospital charges than the non-migration group (65,453.5 vs. 61,525.4 Chinese yuan [CNY]), with a 6% increment (95% CI: 5%−7%; $P<0.001$). The migration group also had a higher incidence of having increased hospital charges (30.89% vs. 22.31%), with an OR of 1.56 (95%CI: 1.48–1.64; $P<0.001$). Propensity score trimming did not change the results substantially.

## Discussion

The present study provided a comprehensive depiction of patient migration for KA in China. Contrary to initial expectations, migration demonstrated limited impact on the postoperative outcomes of KA. In addition, the number of patients that migrated showed a downward trend during the study period.

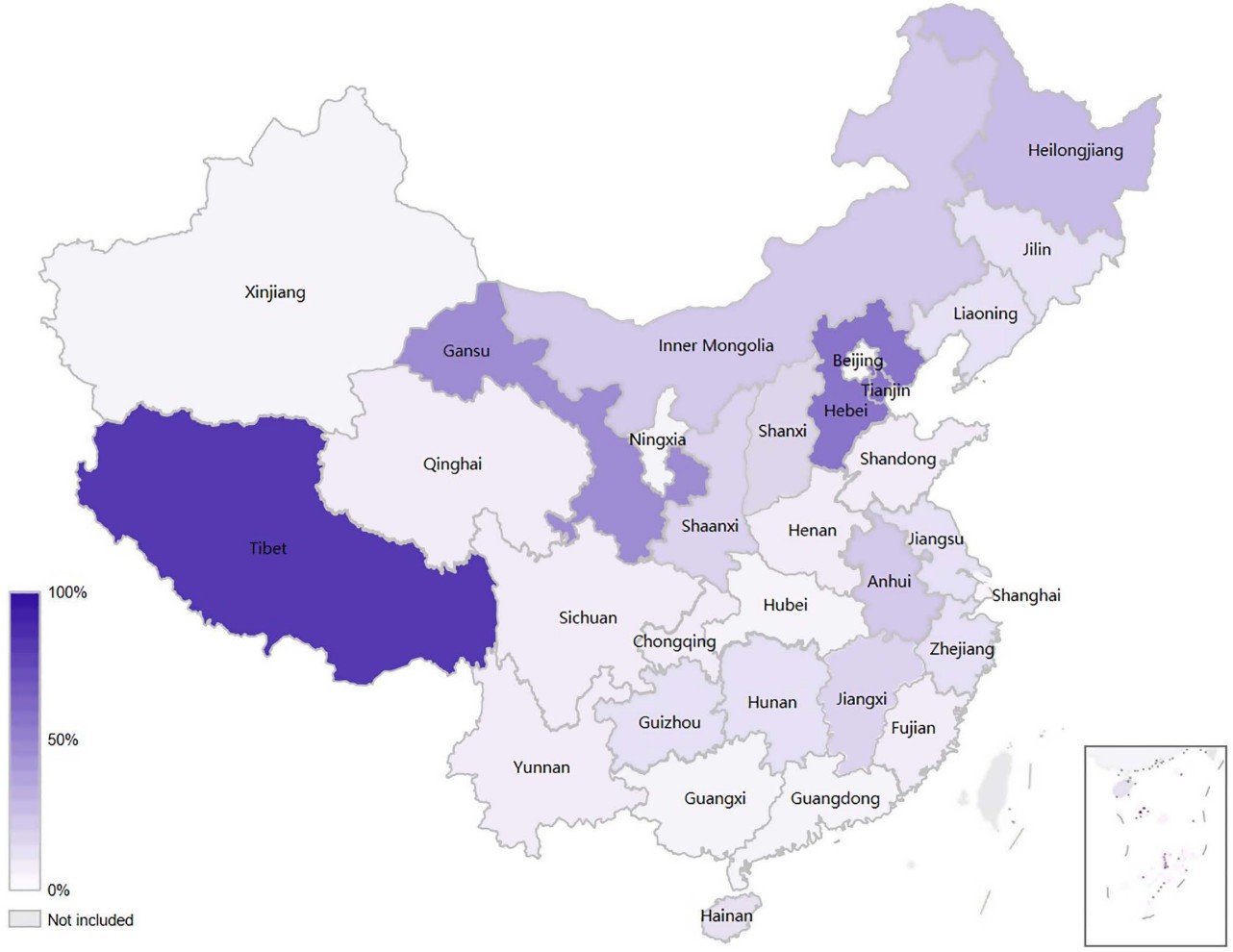

**Fig 3. The percentage of patients who migrated to outside provinces for primary knee arthroplasty in different provinces.**

Trans-provincial medical treatment can be costly. Our study revealed that migrated patients incurred significantly higher hospital charges compared to non-migrants. This cost disparity primarily reflects the premium pricing structures of destination hospitals, particularly high-volume tertiary centers in major metropolitan areas (e.g., Beijing and Shanghai), which typically command higher service fees for comparable procedures [11]. Additionally, institutional factors such as variations in regional pricing policies, and the inclusion of specialized services in treatment packages may contribute to the observed differences. These findings underscore the need for cost-effectiveness evaluations when considering interregional referrals.

In such a situation, a considerable number of patients in China still choose to receive KA outside their provinces. An important challenge in medical care practice today is the disparity between the need for and the availability of specialized healthcare [26]. For some time, the interregional variations in healthcare utilization have often been connected to the lack of accessibility to healthcare in less developed regions [7]. It was found that patients' willingness to migrate for care was mostly derived from their seeking easier accessibility or better quality in medical care [27]. The utilization of KA in China was relatively late compared to that in high-income countries, and KA is still a relatively new and developing surgical procedure [28]. Therefore, a significant proportion of patients tend to go to hospitals located in more developed regions.

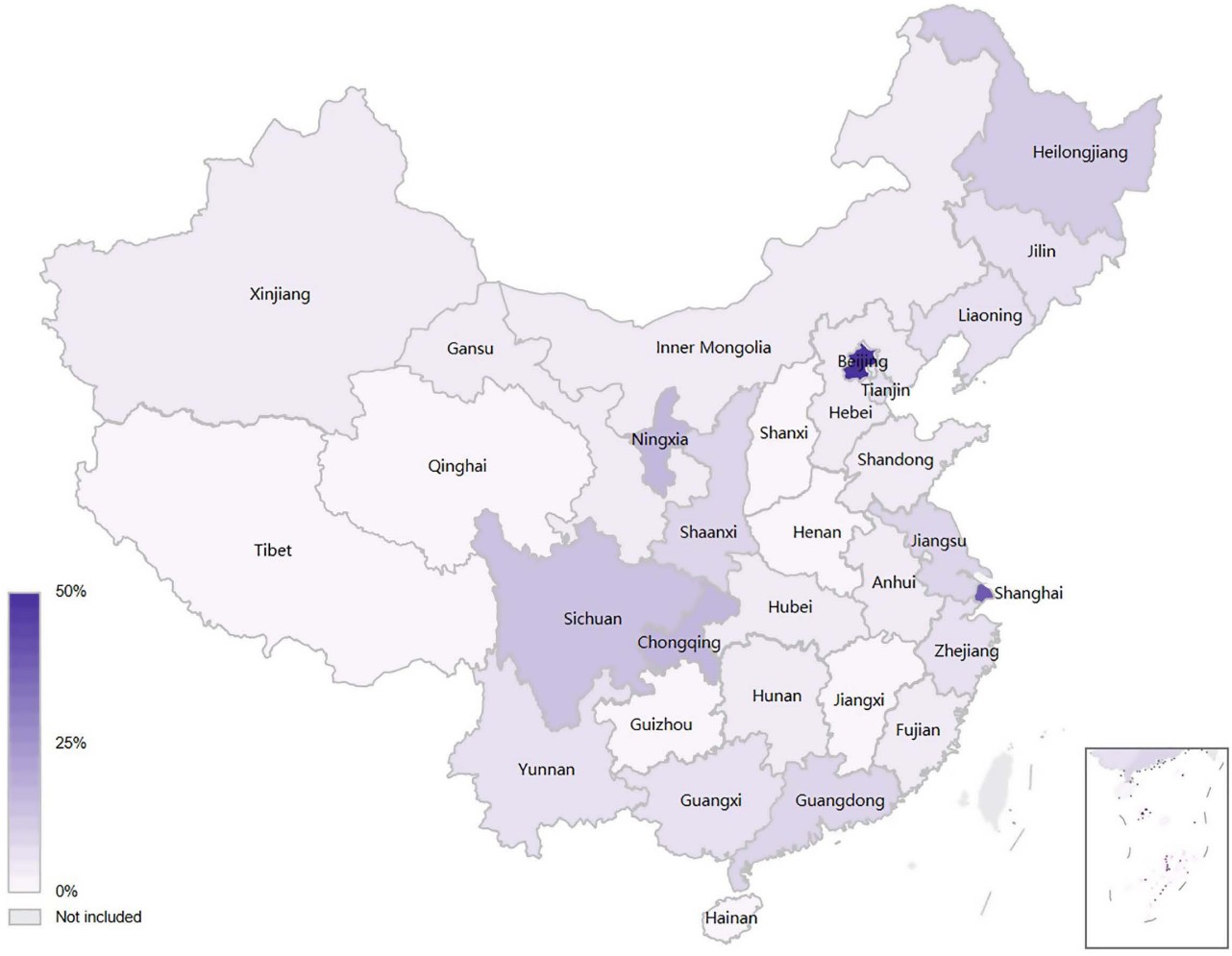

**Fig 4. The percentage of received patients who migrated for primary knee arthroplasty in different provinces.**

However, a downward trend was observed in patient migration for KA. This may reflect several systemic and policy-driven factors. First, China's ongoing healthcare reforms have improved the capacity and quality of regional hospitals, reducing the need for patients to seek care in major urban centers [29]. Second, referential health insurance reimbursement policies, where local treatments receive higher coverage than cross-province care, actively encouraged patients to seek care within their home provinces [30]. Finally, the development of specialist training programs may have enhanced surgical expertise in low-volume hospitals [31]. These efforts aim to reduce regional disparities in healthcare resource distribution.

This study demonstrated that both patient and hospital characteristics have an impact on the patterns of migration. Younger patients were more willing to travel for KA than their counterparts. Generally, young patients have higher activity levels and higher expectations of how long the prosthesis can last [32], which endows them with a stronger driving force to migrate for more prestigious surgeons and advanced techniques. Surprisingly, as the CCI increased, patients were less likely to travel to hospitals located in another region for KA. We speculated that patients with high comorbidity burdens would have lower expectations for activity levels and that they were more vulnerable to the exhaustion caused by

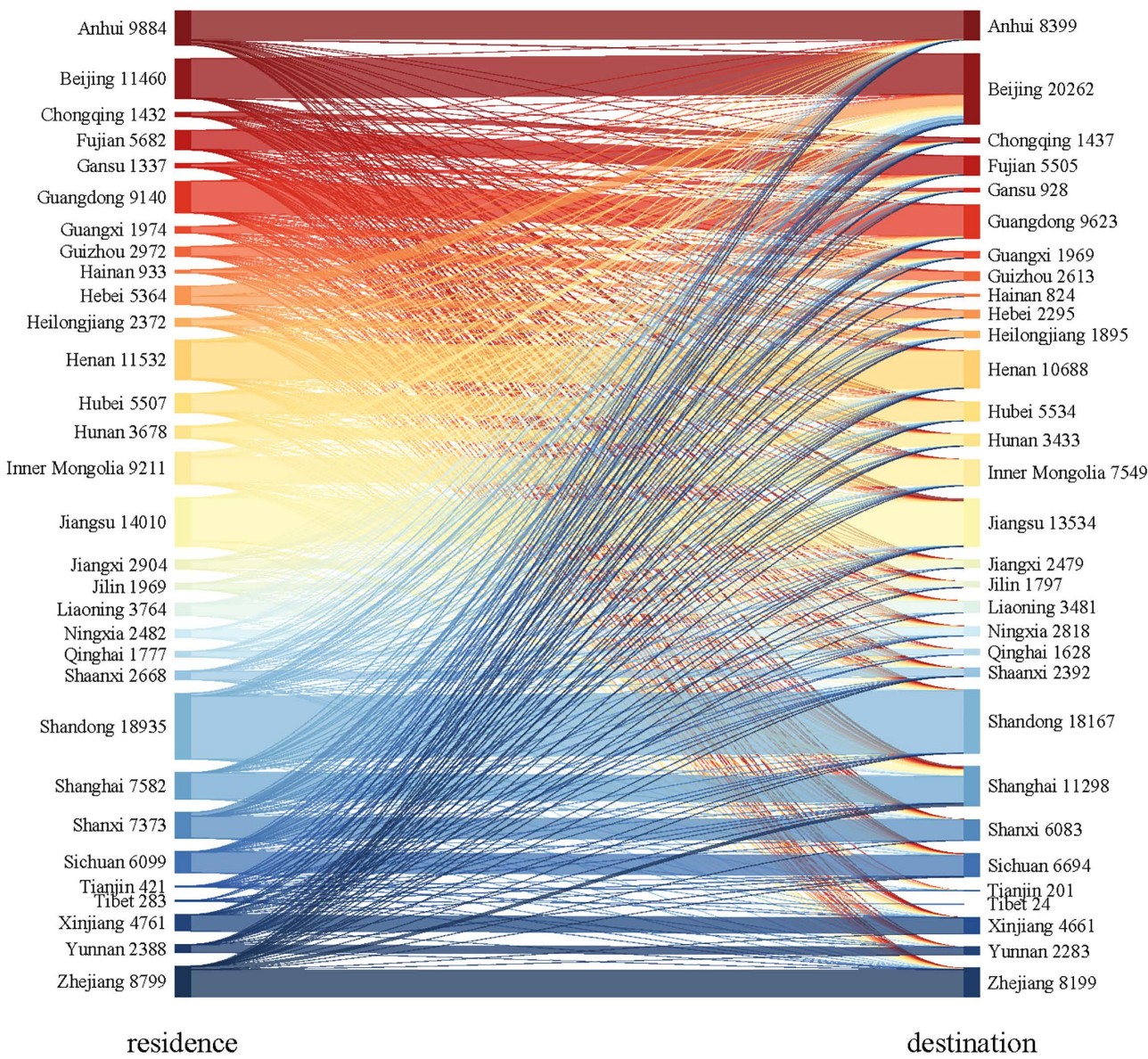

**Fig 5. Sankey diagram of patient migration for primary knee arthroplasty in China.** Sankey diagrams were constructed using the address of the patient's usual residence (first column) and the address of the patients receiving primary knee arthroplasty (second column) as the coordinates before and after the cross-province flow of patients.

long-distance travel. As expected, provincial hospitals and hospitals with high volumes were the favored destinations of migrated patients, as they usually provide more trained surgeons, up-to-date technologies, and comprehensive medical care, which result in better outcomes [33].

The observed mobility reflects the variation in the accessibility of professional joint surgery resources in different areas. Most hospitals in environmentally and economically underdeveloped regions lacked expertise or the resources required [34]. In the current study, we found that Beijing and Shanghai had been the practice centers of KA in China, as the migration rate of patients in their surrounding provinces is high and patients from outside provinces accounted for 44.2% and

**Table 2. Baseline characteristics of propensity score-matched migrated and non-migrated primary knee arthroplasty patients.**

| Characteristics | Non-migrated N = 14,708 | Migrated N = 14,708 | SMD |
|---|---|---|---|
| **Age, mean (SD), years** | 65.7 (8.6) | 65.8 (8.1) | 0.012 |
| **Year (%)** | | | |
| 2013 | 2154 (14.7) | 2154 (14.7) | 0 |
| 2014 | 2406 (16.4) | 2406 (16.4) | 0 |
| 2015 | 2644 (18.0) | 2644 (18.0) | 0 |
| 2016 | 2861 (19.5) | 2861 (19.5) | 0 |
| 2017 | 2467 (16.8) | 2467 (16.8) | 0 |
| 2018 | 2176 (14.8) | 2176 (14.8) | 0 |
| **Sex (%)** | | | 0.009 |
| Female | 11,599 (78.9) | 11,546 (78.5) | |
| Male | 3109 (21.1) | 3162 (21.5) | |
| **Marital status (%)** | | | |
| Married | 13,610 (92.5) | 13,654 (92.8) | 0.011 |
| Single | 798 (5.4) | 752 (5.1) | 0.014 |
| Unknown | 300 (2.0) | 302 (2.1) | 0.007 |
| **Indication (%)** | | | 0.032 |
| Osteoarthritis | 13,714 (93.2) | 13,829 (94.0) | |
| Non-osteoarthritis | 994 (6.8) | 879 (6.0) | |
| **Charlson Comorbidity Index (%)** | | | |
| 0 | 11,536 (78.4) | 11,937 (81.2) | 0.068 |
| 1-2 | 3053 (20.8) | 2670 (18.2) | 0.066 |
| ≥3 | 119 (0.8) | 101 (0.7) | 0.014 |
| **Comorbidity (%)** | | | |
| Myocardial infarction | 16 (0.1) | 13 (0.1) | 0.006 |
| Congestive heart failure | 72 (0.5) | 54 (0.4) | 0.019 |
| Peripheral vascular disease | 216 (1.5) | 198 (1.4) | 0.010 |
| Cerebrovascular disease | 471 (3.2) | 442 (3.0) | 0.011 |
| Dementia | 33 (0.2) | 16 (0.1) | 0.028 |
| Rheumatic disease | 556 (3.8) | 500 (3.4) | 0.020 |
| Peptic ulcer | 58 (0.4) | 57 (0.4) | 0.001 |
| Hemiplegia | 3 (0.0) | 1 (0.0) | 0.012 |
| Chronic renal disease | 42 (0.3) | 24 (0.2) | 0.026 |
| Tumor | 69 (0.5) | 63 (0.4) | 0.006 |
| Liver disease | 309 (2.1) | 271 (1.8) | 0.019 |
| Diabetes | 1700 (11.6) | 1449 (9.9) | 0.055 |
| Coronary artery disease | 299 (2.0) | 242 (1.7) | 0.029 |
| Hypertension | 4737 (32.2) | 4723 (32.1) | 0.002 |
| COPD | 113 (0.8) | 99 (0.7) | 0.011 |
| **Geographical location of residency (%)** | | | |
| North | 5497 (37.4) | 4929 (33.5) | 0.081 |
| East | 5352 (36.4) | 5585 (38.0) | 0.033 |
| North-East | 641 (4.4) | 675 (4.6) | 0.011 |
| South-Central | 1555 (10.6) | 1718 (11.7) | 0.035 |
| South-West | 947 (6.4) | 917 (6.2) | 0.008 |
| North-West | 716 (4.9) | 884 (6.0) | 0.050 |

*(Continued)*

**Table 2.** (Continued)

| Characteristics | Non-migrated<br>N = 14,708 | Migrated<br>N = 14,708 | SMD |
|---|---|---|---|
| **Type of hospital (%)** | | | 0.071 |
| Provincial | 11,577 (78.7) | 11,992 (81.5) | |
| Non-provincial | 3131 (21.3) | 2716 (18.5) | |
| **Hospital volume (%)** | | | |
| >500/year | 7730 (52.6) | 7690 (52.3) | 0.005 |
| 250–500/year | 1977 (13.4) | 2310 (15.7) | 0.064 |
| <250/year | 5001 (34.0) | 4708 (32.0) | 0.042 |

SMD, standardized mean difference; COPD, chronic obstructive pulmonary disease.

**Table 3.** Comparison of postoperative outcomes and financial burden between migrated and non-migrated primary knee arthroplasty patients.

| | Non-migrated<br>N = 14,708 | Migrated<br>N = 14,708 | OR (95% CI) | *P* Value |
|---|---|---|---|---|
| **Postoperative outcomes, No. (%)** | | | | |
| In-hospital mortality | 3 (0.02) | 3 (0.02) | 1.00 (0.20, 4.95) | 1 |
| Pulmonary embolism | 12 (0.08) | 15 (0.10) | 1.25 (0.59, 2.67) | 0.564 |
| Deep vein thrombosis | 169 (1.15) | 183 (1.24) | 1.08 (0.88, 1.34) | 0.453 |
| Wound infection | 20 (0.14) | 22 (0.15) | 1.10 (0.60, 2.02) | 0.758 |
| 30-day readmission | 304 (2.07) | 270 (1.84) | 0.89 (0.75, 1.05) | 0.152 |
| **Financial burden** | | | | |
| Total hospital charges, mean (SD), CNY | 61,525.4 (20,184.2) | 65,453.5 (22,654.0) | 1.06 (1.05, 1.07) | <0.001 |
| Increased total hospital charges (%) | 3282 (22.31) | 4543 (30.89) | 1.56 (1.48, 1.64) | <0.001 |

OR, odds ratio; SD, standard deviation; No., number; CNY, Chinese Yuan.

34.2% of total patients, respectively. The abundant medical resources, development, and convenient transportation in Beijing and Shanghai attract patients from other provinces to migrate for surgery.

A certain scale of cross-district attendance for revision total knee or hip arthroplasty was also observed in the United States. Lawson et al. reported that most patients migrated to medium or large institutions for revision total KA rather than small institutions and to teaching rather than non-teaching institutions [8]. Another study evaluating the effect of hospital size and teaching status for revision total hip arthroplasty on migration patterns also found the same phenomenon [35]. Within the orthopaedic literature, however, the impact of patient migration on postoperative outcomes remains an area of ongoing investigation. Several studies have reported the outcomes of patients migrating for other diseases. Pham et al. reported that treated migrating non-small cell lung cancer patients lived longer than their counterparts [7]. However, in a study conducted by Kohn et al., migrated liver transplantation candidates had a significantly inferior 5-year survival rate compared with non-migrated recipients [9]. In another study conducted on liver transplantation patients in the United States, no differences in post-transplant patient or graft survival between the two groups were found [36]. In this study, migrated patients had a greater financial burden than their counterparts. However, migration did not necessarily bring as many benefits as imagined, especially considering the extra costs and inconvenient reimbursement.

Migrated patients showed no significant difference from their counterparts in terms of in-hospital mortality, wound infection, deep vein thrombosis, and 30-day readmission. Such results suggest that the benefits of migration to destinations well-known to patients are possibly formed by selection bias. As previous studies have shown, migrated patients are generally in better physical condition and have better economic conditions [9,37], which have already been proven to be associated with better postoperative outcomes [38]. Considering the extra costs for travel and accommodations, additional family support, and inconvenient reimbursement, it would be reasonable for clinicians and policy-makers to discuss explicitly whether or not we should discourage the migration of patients in good condition.

The observed migration for KA procedure reflected the inequitable allocation of healthcare resources [34,39]. There is a contradiction between the patients' growing demand for better professional joint surgery healthcare and the unbalanced spatial distribution and accessibility. Priority action should be taken to enhance the facilities, equipment, and personnel in the areas from which patients migrate. Accordingly, China has implemented the construction of national regional medical centers for the purpose of achieving the homogeneity of medical services between regions [40]. Furthermore, the decreasing proportion of migrated patients appears to indicate that the endeavors have started to pay off.

Several strengths of our study are noteworthy. First, this is the first study that investigated patient migration for primary KA. Furthermore, the perspective on patient migration in this study also provided valuable clinical epidemiological data reflecting current KA utilization in China. Finally, we further evaluated the effect of migratory mobility on postoperative outcomes and financial burden. The findings may therefore have important implications for public health policy and healthcare resource allocation, especially in health systems where exist resource maldistribution.

Our study has several limitations. First, due to the inherent limitations of administrative databases, we were not able to assess patients' specific reasons for migration but could only speculate roughly based on patient and hospital characteristics. Second, as a retrospective observational study, unmeasured confounders such as socioeconomic condition might have biased the outcomes. However, the use of a large sample size and a relatively comprehensive analysis of other confounding variables would help mitigate this potential bias. Third, although we observed no tremendous advantages for migrated patients in terms of postoperative complications or financial burden, it is unknown how beneficial migration would be to postoperative pain or functional outcomes, as such information was not recorded in HQMS. Fourth, the current study only assessed short-term outcomes, and there was a lack of long-term follow-up information, such as the quality of life or the longevity of implant. Subsequent studies concentrating on these outcomes are needed to investigate this issue more comprehensively.

## Conclusions

A noticeable number of patients required cross-province migration to get access to KA. After adjusting for patient and hospital characteristics, migration did not significantly impact postoperative outcomes of KA; however, it did result in a greater financial burden. Although the number of patients migrating for KA is trending downward, it is still imperative to take initiatives to reduce disparities in resource allocation instead of relying on cross-district access.

## Author contributions

**Conceptualization:** Fanqiang Meng.

**Data curation:** Huizhong Long.

**Formal analysis:** Fanqiang Meng, Xinjia Deng, Liusong Shen, Huizhong Long.

**Investigation:** Fanqiang Meng, Dongxing Xie.

**Methodology:** Xinjia Deng, Junzhi Sheng, Liusong Shen, Huizhong Long.

**Resources:** Junzhi Sheng, Haoran Feng, Hu Chen.

**Software:** Haoran Feng, Hu Chen.

**Visualization:** Hu Chen, Dongxing Xie.

**Writing – original draft:** Fanqiang Meng, Xinjia Deng.

**Writing – review & editing:** Dongxing Xie, Huizhong Long.

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
