## [Decision Letter · Decision Letter 0]

21 Jul 2025

PONE-D-25-28650
Patterns and impacts of patient migration for primary knee arthroplasty in China: a national retrospective study
PLOS ONE

Dear Dr. Long,

Thank you for submitting your manuscript to PLOS ONE. After careful consideration, we feel that it has merit but does not fully meet PLOS ONE’s publication criteria as it currently stands. Therefore, we invite you to submit a revised version of the manuscript that addresses the points raised during the review process.

We look forward to receiving your revised manuscript.

Kind regards,

Nan Jiang

Academic Editor

PLOS ONE

Journal Requirements: 

2.Thank you for stating the following financial disclosure:

 [This work was supported by the National Key Research and Development Plan (2022YFC3601900 and 2022YFC2505500), the National Natural Science Foundation of China (U21A20352 and 82372474), the China Postdoctoral Science Foundation (2023M743958), the Science and Technology Program of Hunan Province (2025JJ60479 and 2023JJ41003), the Central South University Innovation-Driven Research Programme (2023CXQD031), and the Fundamental Research Funds for the Central Universities of Central South University (2024ZZTS0179 and 2021ZZTS0355).]. 

3. In the online submission form, you indicated that [No data are available. While we are unable to share the data sets, code lists are available on request.].

Reviewers' comments:

Reviewer's Responses to Questions

**Comments to the Author**

1. Is the manuscript technically sound, and do the data support the conclusions?

Reviewer #1: Yes

Reviewer #2: Partly

2. Has the statistical analysis been performed appropriately and rigorously? 

Reviewer #1: I Don't Know

Reviewer #2: Yes

3. Have the authors made all data underlying the findings in their manuscript fully available?

Reviewer #1: Yes

Reviewer #2: Yes

4. Is the manuscript presented in an intelligible fashion and written in standard English?

Reviewer #1: Yes

Reviewer #2: Yes

5. Review Comments to the Author

Reviewer #1: I would like to thank the authors for their great work on a beautiful subjectt.. I will not have any additional suggestions for the articlee..I did not see any harm in publishing it in the journall..

Reviewer #2: Table 2: The table presents a category labeled “non-osteoarthritis” among the indications for knee arthroplasty. Could the authors clarify what diagnoses are included in this group.

Line 171 and Table 3: The manuscript reports increased total hospital charges in the migration group. It would be helpful if the authors elaborate on what this means.

Line 256: The phrase “didn’t make a big difference” in the discussion could be revised for academic tone.

Lines 194 and 257: The observed decline in patient migration from 17.6% in 2013 to 7.6% in 2018 is a notable finding. The authors are encouraged to offer potential explanations for this trend.

Lines 299 onward: The paragraph discusses patient migration in the context of other medical conditions such as liver transplantation and lung cancer. The authors should consider citing current literature on patient migration and outcomes related to other orthopaedic procedures. This would better contextualize their findings within the broader orthopaedic landscape.

6. PLOS authors have the option to publish the peer review history of their article (what does this mean?). If published, this will include your full peer review and any attached files.

Reviewer #1: **Yes: **Mehmet Murat BALA

Reviewer #2: No

---

## [Author Response · Author response to Decision Letter 1]

20 Aug 2025

Reviewer 1

Comment 1: I would like to thank the authors for their great work on a beautiful subject. I will not have any additional suggestions for the article. I did not see any harm in publishing it in the journal.

Response: We sincerely appreciate the reviewer’s positive evaluation of our work. We are grateful for the time and consideration given to our manuscript.

Reviewer 2

Comment 1: Table 2: The table presents a category labeled “non-osteoarthritis” among the indications for knee arthroplasty. Could the authors clarify what diagnoses are included in this group.

Response: Thank you for your insightful comment. We have clarified the composition of the “non-osteoarthritis” category in the Methods section. Specifically, the indications for knee arthroplasty (KA) were divided into two groups: osteoarthritis and non-osteoarthritis. The non-osteoarthritis category included conditions such as rheumatoid arthritis, osteonecrosis, and other knee disorders.

Action: “We divided the indications for KA into two categories: osteoarthritis and non-osteoarthritis (e.g., rheumatoid arthritis, osteonecrosis, and other knee disorders)[15].” (Page 8, lines 146-148 in the revised manuscript)

Comment 2: Line 171 and Table 3: The manuscript reports increased total hospital charges in the migration group. It would be helpful if the authors elaborate on what this means.

Response: We appreciate the reviewer’s valuable suggestion. Patients who migrated for KA faced increased total hospital charges, attributable to higher fee structures at destination medical centers, particularly tertiary hospitals in major cities, and potential differences in treatment protocols and service intensity between regions. Further details have been added to the Discussion section.

Action: “Trans-provincial medical treatment can be costly. Our study revealed that migrated patients incurred significantly higher hospital charges compared to non-migrants. This cost disparity primarily reflects the premium pricing structures of destination hospitals, particularly high-volume tertiary centers in major metropolitan areas (e.g., Beijing and Shanghai), which typically command higher service fees for comparable procedures [11]. Additionally, institutional factors such as variations in regional pricing policies, and the inclusion of specialized services in treatment packages may contribute to the observed differences. These findings underscore the need for cost-effectiveness evaluations when considering interregional referrals.” (Page 20, lines 262-270 in the revised manuscript)

Comment 3: Line 256: The phrase “didn’t make a big difference” in the discussion could be revised for academic tone.

Response: We appreciate the reviewer’s valuable suggestion and have revised the phrase “didn’t make a big difference” to improve the academic tone.

Action: “Contrary to initial expectations, migration demonstrated limited impact on the postoperative outcomes of KA.” (Page 20, lines 258-259 in the revised manuscript)

Comment 4: Lines 194 and 257: The observed decline in patient migration from 17.6% in 2013 to 7.6% in 2018 is a notable finding. The authors are encouraged to offer potential explanations for this trend.

Response: Thank you for your constructive suggestion. The decline in migration rates primarily reflects China’s healthcare system improvements, including enhanced regional hospital capacity through healthcare reforms, preferential health insurance reimbursement policies that incentivize local care, and specialized training elevating surgical expertise in low-volume hospitals. These policy interventions may make quality KA care more accessible locally. The potential explanations were detailed in the Discussion section.

Action: “However, a downward trend was observed in patient migration for KA. This may reflect several systemic and policy-driven factors. First, China’s ongoing healthcare reforms have improved the capacity and quality of regional hospitals, reducing the need for patients to seek care in major urban centers[30]. Second, referential health insurance reimbursement policies, where local treatments receive higher coverage than cross-province care, actively encouraged patients to seek care within their home provinces[31]. Finally, the development of specialist training programs may have enhanced surgical expertise in low-volume hospitals[32]. These efforts aim to reduce regional disparities in healthcare resource distribution.” (Page 21, lines 296-304 in the revised manuscript)

Add three new references:

30. Yip W, Fu H, Chen AT, Zhai T, Jian W, Xu R, et al. 10 years of health-care reform in China: progress and gaps in Universal Health Coverage. Lancet. 2019;394(10204):1192-204. doi: 10.1016/s0140-6736(19)32136-1

31. Han J, Meng Y. Institutional differences and geographical disparity: the impact of medical insurance on the equity of health services utilization by the floating elderly population - evidence from China. Int J Equity Health. 2019;18(1):91. doi: 10.1186/s12939-019-0998-y

32. Chen Z. Launch of the health-care reform plan in China. Lancet. 2009;373(9672):1322-4. doi: 10.1016/s0140-6736(09)60753-4

Comment 5: Lines 299 onward: The paragraph discusses patient migration in the context of other medical conditions such as liver transplantation and lung cancer. The authors should consider citing current literature on patient migration and outcomes related to other orthopaedic procedures. This would better contextualize their findings within the broader orthopaedic landscape.

Response: Thank you for this valuable suggestion. After a thorough literature review, we have expanded our discussion on patient migration by incorporating current evidence specific to orthopaedic procedures, such as revision total knee and hip arthroplasty. However, we explicitly noted that the impact of migration on postoperative outcomes in orthopaedics remains an active area of investigation, bridging this gap with existing literature on other medical conditions (e.g., lung cancer and liver transplantation). This addition strengthens the clinical relevance of our findings while acknowledging the need for further orthopaedic-specific research.

Action: “A certain scale of cross-district attendance for revision total knee or hip arthroplasty was also observed in the United States. Lawson et al. reported that most patients migrated to medium or large institutions for revision total KA rather than small institutions and to teaching rather than non-teaching institutions[8]. Another study evaluating the effect of hospital size and teaching status for revision total hip arthroplasty on migration patterns also found the same phenomenon[36]. Within the orthopaedic literature, however, the impact of patient migration on postoperative outcomes remains an area of ongoing investigation. Several studies have reported the outcomes of patients migrating for other diseases. Pham et al. reported that treated migrating non-small cell lung cancer patients lived longer than their counterparts [7]. However, in a study conducted by Kohn et al., migrated liver transplantation candidates had a significantly inferior 5-year survival rate compared with non-migrated recipients [9]. In another study conducted on liver transplantation patients in the United States, no differences in post-transplant patient or graft survival between the two groups were found [37]. In this study, migrated patients had a greater financial burden than their counterparts. However, migration did not necessarily bring as many benefits as imagined, especially considering the extra costs and inconvenient reimbursement.” (Pages 22-23, lines 329-346 in the revised manuscript)

Add one new reference:

36. Illgen RL, Lewallen DG, Yep PJ, Mullen KJ, Bozic KJ. Migration Patterns for Revision Total Hip Arthroplasty in the United States as Reported in the American Joint Replacement Registry. J Arthroplasty. 2021;36(4):1401-6. doi: 10.1016/j.arth.2020.10.030

---

## [Decision Letter · Decision Letter 1]

1 Sep 2025

Patterns and impacts of patient migration for primary knee arthroplasty in China: a national retrospective study

PONE-D-25-28650R1

Dear Dr. Long,

We’re pleased to inform you that your manuscript has been judged scientifically suitable for publication and will be formally accepted for publication once it meets all outstanding technical requirements.

Kind regards,

Nan Jiang

Academic Editor

PLOS ONE

Reviewer's Responses to Questions

**Comments to the Author**

1. If the authors have adequately addressed your comments raised in a previous round of review and you feel that this manuscript is now acceptable for publication, you may indicate that here to bypass the “Comments to the Author” section, enter your conflict of interest statement in the “Confidential to Editor” section, and submit your "Accept" recommendation.

Reviewer #2: All comments have been addressed

2. Is the manuscript technically sound, and do the data support the conclusions?

Reviewer #2: Yes

3. Has the statistical analysis been performed appropriately and rigorously? 

Reviewer #2: Yes

4. Have the authors made all data underlying the findings in their manuscript fully available?

Reviewer #2: Yes

5. Is the manuscript presented in an intelligible fashion and written in standard English?

Reviewer #2: Yes

6. Review Comments to the Author

Reviewer #2: Thank you for addressing the previous comments and for your work on the manuscript. I have no further suggestions, and I believe the article is now well prepared and suitable for publication.

7. PLOS authors have the option to publish the peer review history of their article (what does this mean?). If published, this will include your full peer review and any attached files.

Reviewer #2: No

---

## [Editor Report · Acceptance letter]

PONE-D-25-28650R1

PLOS ONE

Dear Dr. Long,

I'm pleased to inform you that your manuscript has been deemed suitable for publication in PLOS ONE. Congratulations! Your manuscript is now being handed over to our production team.

Kind regards,

on behalf of

Dr. Nan Jiang

Academic Editor

PLOS ONE